# Research

evolution, genomics

dosage compensation, sexual antagonism, sex chromosomes, *Teleogryllus oceanicus*

**Author for correspondence:**
Jack G. Rayner
e-mail: jr228@st-andrews.ac.uk

# Variable dosage compensation is associated with female consequences of an X-linked, male-beneficial mutation

Jack G. Rayner, Thomas J. Hitchcock and Nathan W. Bailey

Centre for Biological Diversity, University of St Andrews, St Andrews KY16 9TH, UK

JGR, 0000-0001-9259-9046; TJH, 0000-0002-6378-5023; NWB, 0000-0003-3531-7756

Recent theory has suggested that dosage compensation mediates sexual antagonism over X-linked genes. This process relies on the assumption that dosage compensation scales phenotypic effects between the sexes, which is largely untested. We evaluated this by quantifying transcriptome variation associated with a recently arisen, male-beneficial, X-linked mutation across tissues of the field cricket *Teleogryllus oceanicus*, and testing the relationship between the completeness of dosage compensation and female phenotypic effects at the level of gene expression. Dosage compensation in *T. oceanicus* was variable across tissues but usually incomplete, such that relative expression of X-linked genes was typically greater in females. Supporting the assumption that dosage compensation scales phenotypic effects between the sexes, we found tissues with incomplete dosage compensation tended to show female-skewed effects of the X-linked allele. In gonads, where expression of X-linked genes was most strongly female-biased, ovaries-limited genes were much more likely to be X-linked than were testes-limited genes, supporting the view that incomplete dosage compensation favours feminization of the X. Our results support the expectation that sex chromosome dosage compensation scales phenotypic effects of X-linked genes between sexes, substantiating a key assumption underlying the theoretical role of dosage compensation in determining the dynamics of sexual antagonism on the X.

## 1. Introduction

The X chromosome is widely predicted to be a hotspot for genes with sexually antagonistic fitness effects in XX/XY and XX/XO systems [1–3]. The role of allelic dominance in mediating these effects has been well studied, indicating sexual antagonism on the X should usually favour the spread of female-beneficial variants (as females transmit twice as many copies to the next generation), unless recessive, in which case male-beneficial variants can more readily invade [4–7]. These predictions could explain why genes with male-biased expression often appear underrepresented on X chromosomes [8–11]. However, a less commonly considered factor affecting predictions of sexual antagonism on sex chromosomes is dosage compensation (i.e. whether males and females exhibit differences in expression of X-linked relative to autosomal genes) [12,13]. Recent theory has shown that the presence and completeness of dosage compensation are likely to play an important role in mediating sexual antagonism on the X [7,14], but the assumptions underlying this hypothetical role have rarely been addressed in empirical work.

If the sexes are equal in their expression of X-linked genes relative to autosomal genes—for example, through upregulation in heterogametic males [15]—then phenotypic consequences of X-linked alleles are expected to be of similar magnitude in hemizygous males and homozygous females [4,7,14]. This view has empirical support from gene translocation experiments in species with complete or near-complete dosage compensation [16,17]. By contrast, if there is the lower relative expression of X-linked genes in males, such as in species where

dosage compensation is absent or incomplete [18], then X-linked variants will tend to have greater phenotypic effects and thus more prominent fitness consequences in homozygous females [7]. In this case, X-linked variants are more likely to experience selection favouring female fitness [19]. Thus, the likelihood of alleles with sexually antagonistic fitness effects spreading is predicted to be affected by the completeness of dosage compensation (Box 1). An important first step testing this idea is to establish whether sex differences in phenotypic effects scale with the extent of dosage compensation.

Opportunities to test this expectation are rare, because most phenotypes have complex, polygenic architectures [6] and allelic variants with sex-specific fitness effects are difficult to detect. We capitalized on a system of Hawaiian field crickets (*Teleogryllus oceanicus*) in which adaptive male song loss has recently evolved, providing the opportunity to test how effects of an X-linked mutation with sex-specific fitness consequences relate to patterns of dosage compensation. In *T. oceanicus*, adaptive silence is caused by altered male wing venation (the 'flatwing' phenotype—other silencing phenotypes have also been observed [20]), which precludes the production of the song. On the island of Kauai, the development of this male phenotype is caused by an X-linked allele, *flatwing* [21]. Song loss protects males from an acoustically orienting parasitoid fly whose larvae are lethal endoparasites of *T. oceanicus* adults [22], whereas females do not have differentiated wings and cannot sing.

The sex determination system of *T. oceanicus* is XO, so males carry one copy of the X chromosome (and thus *flatwing* locus) and females two, but the *flatwing* allele does not have obvious phenotypic consequences for female wing morphology. One might therefore expect *flatwing* to have little if any effect on female gene expression or associated phenotypes; however, this does not appear to be the case. Recent reports indicate that the *flatwing* mutation has pleiotropic or otherwise linked consequences for female gene expression [23], and for female life-history traits (reduced reproductive investment, increased rate of mating failure, increased somatic mass index, growth rate) [24–26]. While there is therefore evidence that *flatwing* has sexually antagonistic fitness effects in at least some contexts, such as reproductive investment, we can confidently infer only that it is under strong sex-biased selection, providing considerable fitness benefits to males via sex-limited phenotypic effects on wing morphology (Zuk *et al.* [22] found < 1% of flatwing males harboured lethal endoparasitic larvae, versus > 30% of normal-wing males). Female fitness effects are likely to be minor in comparison. A potential explanation, then, for the surprising magnitude of *flatwing*-associated gene expression effects in females is that incomplete dosage compensation causes greater female expression of X-linked genes. In the case of *flatwing*, this would be unlikely to have impeded its spread, due to dramatic fitness benefits in males. However, in the context of less strongly selected X-linked alleles with sexually antagonistic fitness effects, such a role for dosage compensation could have an important influence by increasing the magnitude of phenotypic effect in females relative to males. Given its location on the X-chromosome and male-specific fitness benefits, *flatwing* affords a useful opportunity to test this role.

We used multiple RNA-seq datasets to measure gene expression in *normal-wing* and *flatwing* genotypes, within each sex and across five tissues. By comparing gene expression effects of the *flatwing* allele in different sexes, we could test for

differences in its phenotypic effects at the transcriptome level. Specifically, by quantifying differences in expression between morph genotypes within each sex, and the relative expression of X-linked to autosomal genes, we were able to test the presumed association between female expression effects of a male-beneficial X-linked allele, and the completeness of dosage compensation. Based on previous findings of large gene expression effects of *flatwing* in female gonad and somatic tissues [23,24], we predicted that XX females would show greater relative expression of X-linked genes compared with XO males (i.e. incomplete dosage compensation prediction 1), though the extent would likely vary across tissues [11,12]. Next, we predicted that tissues exhibiting greater female-skew in gene expression effects of carrying the *flatwing* allele would be those with less complete dosage compensation (prediction 2). Such a pattern would support the interpretation that substantial female consequences of the male-beneficial *flatwing* allele are a consequence of incomplete dosage compensation, and that this should be an important parameter in predicting the spread of sexually antagonistic variants. Given the theoretical relationship between dosage compensation and spread of sexually antagonistic variants (box 1), our final prediction was that tissues in which dosage compensation was least evident (i.e. female-biased expression of X-linked genes most pronounced) would exhibit feminization of the X-chromosome [9]. Following our results from prediction 1, which reported little or no evidence of dosage compensation in sexually dimorphic gonads, we tested this by asking whether ovaries-limited genes were overrepresented on the X (prediction 3).

## 2. Methods

### (a) RNA-seq datasets

We analysed previously published RNA-seq data collected from pure-breeding *normal-wing* and *flatwing* cricket lines across a range of tissues: neural tissue at 7 days post-adulthood ($n = 48$ libraries, 6 replicates per sex × genotype) [23]; neural, thoracic and gonad tissue at *ca* 14 days post-adulthood ($n = 36$ libraries, 3 replicates per tissue × sex × genotype) [24]; and developing wingbuds ($n = 12$ libraries, 3 replicates per sex × genotype) [27]. All cricket populations used in the above studies were derived from a single sample of a wild population in Wailua, Kauai ([25], electronic supplementary material). All males were hemizygous for their respective morph genotype (*flatwing* versus *normal-wing*), while females were homozygous. Flatwing and normal-wing lines from the individual studies also shared recent ancestry, having been derived from mixed laboratory populations, so replicate lines should on average differ at the causative *flatwing* locus/loci, and closely linked loci. The use of replicate lines of each morph enabled us to associate differences in gene expression with differences in morph genotype and minimize the possibility of experimental confounds due to differing background effects [23,24,28].

As well as being from different experimental conditions, neural data from [23] and [24] were collected at different ages (7d and approx. 14d post-adult eclosion), so were treated as different samples, henceforth designated *neural_7d* and *neural_14d*, respectively. In the study from which *neural_7d* samples were collected [23], crickets had also been subjected to different social acoustic regimes during rearing, which interacted significantly with sex and morph in affecting gene expression. These acoustic regimes included a 'silent' treatment, in which conspecific song is absent (as in the wild Kauai population), and a 'song' treatment where crickets were exposed to high levels of playback of normal-wing

**Box 1.** The role of dosage compensation in sexual antagonism.

Here, we illustrate the hypothetical contribution of dosage compensation to patterns of sexual antagonism associated with an additive, X-linked allele, based on existing theory [7,14].

(*a*) Describes an XO system of sex determination, in which males carry one copy of the X and females two; similar dynamics are expected with XX/XY systems of sex determination. With respect to genotype at a given X-linked locus, males carry one copy of an additive allele (here, *a*) whereas females carry two (*aa*). Colours indicate chromosome parental origin (blue = paternal, pink = maternal). There is a 2 : 1 greater maternal contribution of X chromosomes to the next generation.

(*b*) Shows how differences in the phenotypic effects of carrying *a* or *aa* genotypes are expected to be affected by global differences in expression of X-linked loci (i.e. chromosome-wide differences in expression affecting all genes on the X more or less equivalently, or *dosage compensation*). Here, $\delta$ represents the magnitude of phenotypic consequence of the specific locus in *aa* females relative to *a* males, which is influenced by the extent of dosage compensation. Three plausible scenarios are illustrated: complete dosage compensation ($\delta = 1$), incomplete dosage compensation ($1 > \delta < 2$; for illustration $\delta = 1.5$), and no dosage compensation ($\delta = 2$). We illustrate dosage compensation occuring via upregulation of the X in males, though this could also occur through downregulation or X-inactivation in females. The less complete dosage compensation is (i.e. as $\delta \to 2$), the greater the expected phenotypic consequences in homozygous females of the additive allele, *a*, relative to hemizygous males. Therefore, if all else is equal, X-linked alleles should be under female-biased selection when dosage compensation is absent or incomplete.

(*c*) Then illustrates how $\delta$ is expected to influence the likelihood that the *a* allele will invade the X chromosome [7,14]. Here, *f* is the marginal benefit to cost ratio of *aa* females relative to *a* males, with fitness effects equal when $f = 1$. Blue and pink shaded regions indicate conditions under which male- and female-beneficial antagonistic variants are more likely to invade, respectively. If dosage compensation is incomplete, then $\delta > 1$, and selection will tend to favour female-beneficial variants. Note that our study primarily concerns the patterns in (*a*) and (*b*); (*c*) is included to illustrate how these patterns are predicted to influence evolutionary dynamics of sexual antagonism.

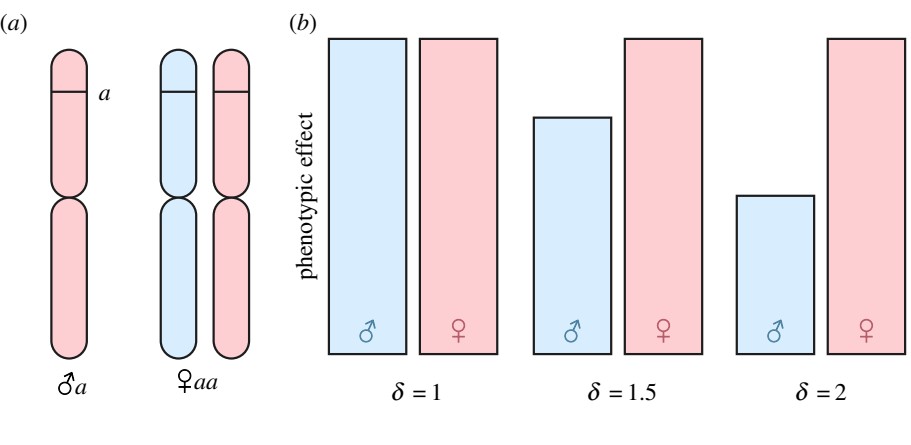

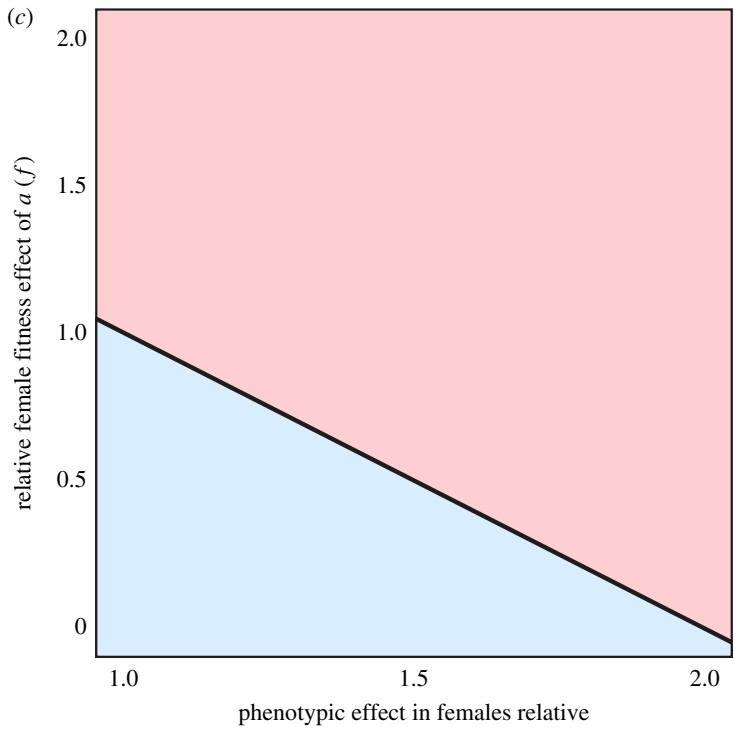

male *T. oceanicus* song models meant to mimic a population dense with calling males. For our purposes of calculating 'baseline' differences in gene expression between normal-wing (*Nw*) and flatwing (*Fw*) genotypes, specifically addressing the pattern of greater effects in females, we used RNA-seq data from the silent treatment. Details of RNA extraction and sequencing procedures are given in the respective publications, and all RNA-seq data is publicly available (electronic supplementary material, table S1).

## (b) Alignment and quantification of RNA-seq data

Trimmed RNA libraries from each of the samples were aligned to the *T. oceanicus* genome assembled by [29] using HISAT2 [30], then individual transcriptome assemblies were generated using Stringtie [31], restricting transcript quantification to the annotated gene set ($n = 19\,157$) provided by [29]. For each dataset, a single reference transcriptome was created by merging individual transcriptomes, used by Stringtie to quantify individual gene expression for consensus transcripts. Estimated gene counts were used as input in R for further analysis, and TMM normalization of library counts performed for libraries from each of the tissues [32].

## (c) Quantifying differential gene expression associated with flatwing

Differential expression analysis was performed in the EdgeR package in R v. 3.4.1 [32]. After filtering genes not expressed at greater than 1 count per million in at least three samples for each tissue, differential gene expression between *Nw* and *Fw* lines was analysed by constructing negative binomial generalized linear models for each tissue separately and performing pairwise contrasts for 'genotype' (i.e. *Nw – Fw*) in each sex. Significance was tested using likelihood ratio tests, with an initial false discovery rate adjusted significance threshold of FDR < 0.05, and corroborating our results using a more robust threshold of FDR < 0.01. This more stringent criterion was used to check that the heightened likelihood of false positives in datasets with fewer replicates per group did not influence our results, and that results were consistent across different significant thresholds. To further check that differences in replicate size, which was greater in the neural_7d tissue dataset than in others, did not affect our results, we iteratively subsampled three libraries from each of the six replicates for tissue × sex × genotype in this dataset and evaluated whether results were consistent (we found that they were; see electronic supplementary material).

Identification of differentially expressed (DE) genes is influenced by variance among samples within groups (which should be smaller than the variance between groups), thus fewer differentially expressed genes might be reported between groups of samples with greater within-group variance. Although we expected that variance should be similar across genotypes and sexes within each dataset, we calculated the biological coefficient of variance (i.e. variance within morph genotypes; calculated as the square root of common dispersion [32]) for each sex-by-tissue combination, to check this did not skew our results. The results showed no trend for lower variance in females (electronic supplementary material, table S2).

Our primary focus in testing the effect of the *flatwing* locus was on the number of DE genes. However, the magnitude of expression differences among DE genes is an important factor to take into account, so we also summed absolute $\log_2$-fold changes across all differentially expressed genes, and visually compared this across sexes and tissues as an approximation of total effect on gene expression irrespective of the genes that were expressed. We did not perform statistical comparisons between summed log-fold changes, as this would be confounded by differences in the contribution of individual genes. Instead, this approach helped ensure

that comparing numbers of DE genes did not inadvertently mask contrasting variation in the magnitude of fold changes.

## (d) Quantifying dosage compensation of X-linked genes

Two common approaches to test for dosage compensation in RNA-seq data are (i) to statistically compare differences in expression of X and autosomal genes for each sex separately, then contrast differences between sexes (requiring two comparisons) and (ii) to compare female:male (F : M) expression ratios between X and autosomal genes [12]. Because we found X-linked genes were consistently expressed more highly than autosomal genes when pooled across linkage groups (electronic supplementary material, figures S1 and S2), which complicated comparison of X : A expression ratios (as females did not exhibit a 1 : 1 ratio for null comparison), we used the approach of comparing F : M ratios between X and autosomal genes to quantify dosage compensation (as autosomal genes showed an F : M expression ratio approximately centred around one). This also more directly addresses our hypothesis that X-linked genes are more highly expressed in females. Note, however, that the two approaches produced results with similar interpretation (electronic supplementary material, figure S1). In calculating F : M expression ratios across genes, we first normalized gene counts, then excluded zero counts and genes expressed in just one sex. After averaging expression across replicates within each sex, F : M expression ratios were compared between X and autosomal genes using Wilcoxon rank-sum tests. We found no indication that relative expression of X-linked to autosomal genes differed between *flatwing*/*normal-wing* genotypes of each sex. We therefore pooled normal-wing and flatwing samples within each sex when calculating differences between tissues in the degree of dosage compensation.

## (e) Testing feminization of the X in the gonads

The above analyses revealed greater relative expression of X-linked genes in female ovaries with respect to male testes, so we tested whether putatively female-beneficial genes—those with ovaries-limited expression—were disproportionately X-linked. Such a pattern could support the view that female-beneficial variants on the X chromosome are favoured when dosage compensation is absent or incomplete. We defined genes with sex-limited gonad expression as those expressed greater than 1 count per million in all six samples of one sex, and less than 1 count per million in all six samples of the opposite sex. Too few genes showed sex-limited expression in somatic tissues to perform the same comparison.

# 3. Results

## (a) Expression levels of X-linked genes were female-biased but varied across tissues

We found variable degrees of incomplete dosage compensation across tissues and morph genotypes of *T. oceanicus*. Relative expression of X-linked genes was typically greater in females than males, supporting our first prediction (figure 1*a*). *T. oceanicus* thus represents another of a growing list of species which show incomplete sex chromosome dosage compensation [18]. However, this pattern varied across tissues, with sexually dimorphic gonads showing the most extensive female-biased expression (with the median expression of X-linked genes 1.98 times greater in females; figure 1*a*). This is consistent with previous work suggesting differences in expression of X-linked genes are strongly exaggerated in gonads, with dosage compensating mechanisms often appearing absent [12,33]. Neural tissue at 7d, in contrast, did not show significant sex differences

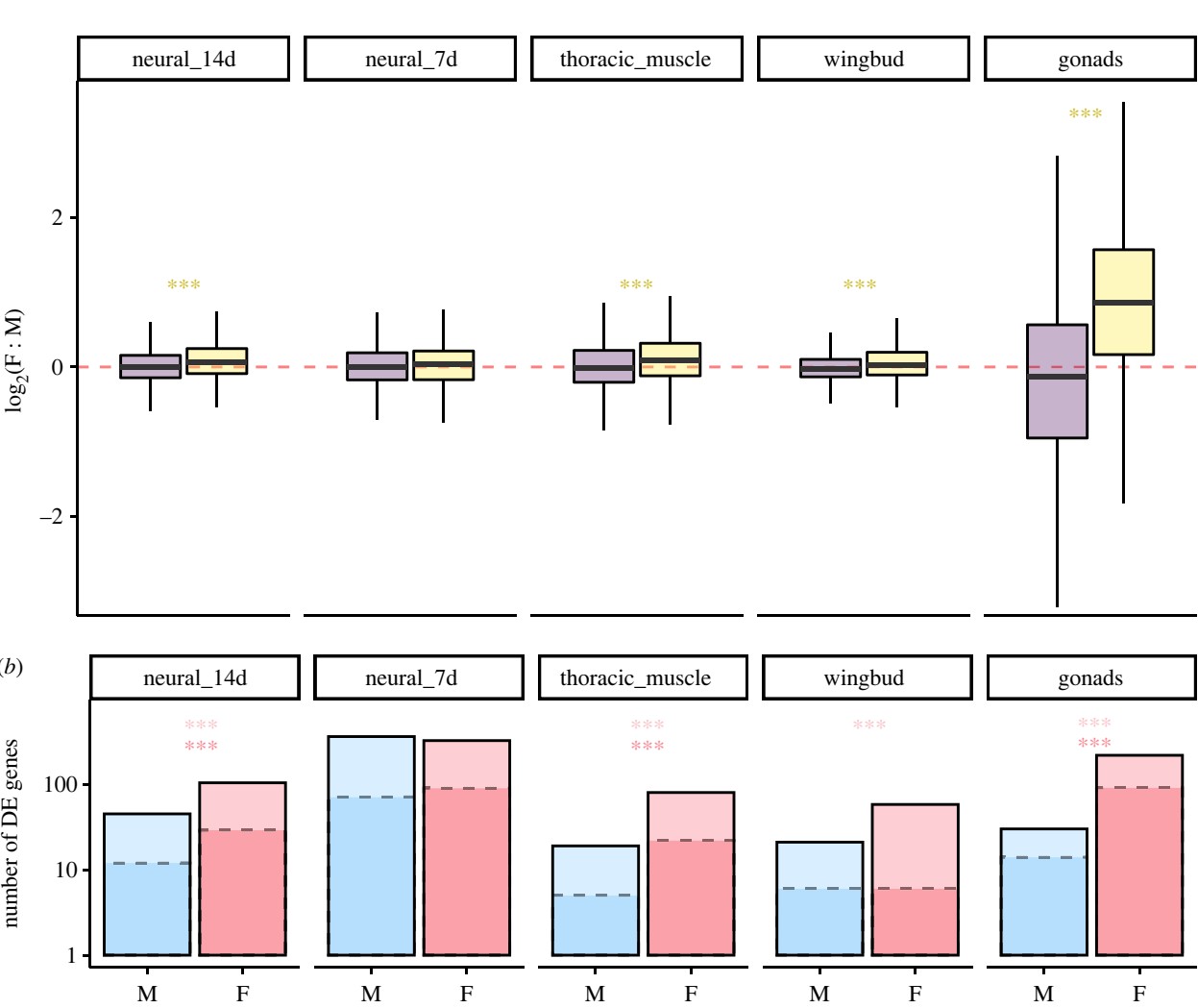

**Figure 1.** Variable dosage compensation and differential expression associated with *flatwing* across *T. oceanicus* tissues. (a) Females showed greater expression (log$_2$(F : M) > 0) of X-linked relative to autosomal genes in all tissues but neural tissue at 7 days. Asterisks denote significantly female-biased expression of X-linked relative to autosomal genes, from Wilcoxon rank-sum tests. Boxplots show medians and interquartile range, with outliers not shown. The dashed red line illustrates the null expectation of equal expression. (b) Females also showed a greater number of DE genes between *flatwing* and *normal-wing* genotypes than males, in all but neural tissue at 7d. Dashed lines and darker shading within bars illustrate the number of DE genes at FDR < 0.01, while the full height of the bars illustrates DE gene numbers at FDR < 0.05. Light red asterisks denote differences between sexes among genes DE at FDR < 0.05, darker red asterisks those DE at FDR < 0.01, from Pearson's chi-squared tests. Note the log$_{10}$ y-axis scale. **$p$ < 0.01, ***$p$ < 0.001.

in expression of X-linked relative to autosomal genes (figure 1a). The remaining somatic tissues showed patterns of female-biased X-dosage, albeit much smaller than observed in gonads, with median X-expression 1.05 times that of males (figure 1a). The X-chromosome also showed a general trend for heightened expression compared with autosomes (electronic supplementary material, figure S1), due to the low expression of genes in some autosomal linkage groups (electronic supplementary material, figure S2). As in previous studies [34,35], we found that the magnitude of female-biased X-expression tended to increase with greater average expression level in tissues showing incomplete dosage compensation (electronic supplementary material, figure S3).

## (b) Female-biased gene expression effects of flatwing

Females tended to show a greater number of DE genes between morph genotypes compared with males. This large apparent effect of *flatwing* in females is consistent with

previous results and analyses and was apparent in all tissues but neural tissue at 7d in which there was no apparent difference. In wingbuds, the above pattern was apparent for DE genes with an FDR < 0.05, but not among those DE at the more stringent FDR < 0.01 threshold (figure 1b). A possible explanation for this incongruity is that there were relatively few DE genes in this tissue. Summed absolute log-fold changes across all DE genes showed a similar pattern of female-biased effect (electronic supplementary material figure S4). Across both sexes, a large majority (n = 1251 of 1407 located in the linkage map) of genes DE between morph genotypes were autosomal, with X-linked genes showing no disproportionate overrepresentation compared with autosomal genes ($\chi^2_1 = 2.475$, $p = 0.116$). However, the pattern of large gene expression effects in females was nevertheless evident among these X-linked genes, and marginally heightened relative to that of autosomes (electronic supplementary material, figure S5). Though local patterns of dosage compensation are difficult to infer given the

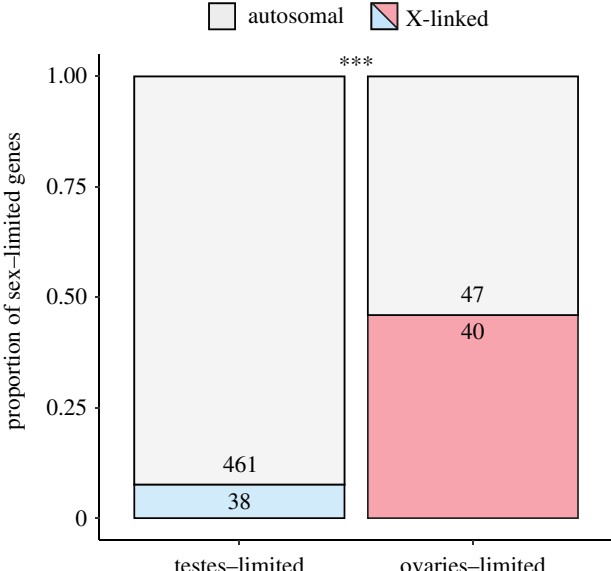

**Figure 2.** Female ovary-specific genes were much more likely to be X-linked compared with testes-specific genes. Numbers within bar segments indicate the total number of genes in each category, and asterisks indicate significant ($p < 0.001$) differences in proportions of X-linked genes.

presence of genes with sex-biased expression, female-biased expression appeared relatively evenly distributed across the X (electronic supplementary material, figure S6).

Next, we found that the female-biased expression of X-linked genes was positively associated with the magnitude of female-biased differential expression between *flatwing* and *normal-wing* lines, a pattern that is in line with our second prediction. Across the five tissues, female-biased X-dosage was positively correlated with female-biased expression effects of carrying *flatwing*, and the relationship approached statistical significance at $p < 0.05$ (Spearman's rho = 0.9, $p = 0.083$ at both FDR < 0.05 and FDR < 0.01 DE thresholds; electronic supplementary material, figure S7). This apparent association was most strikingly illustrated by the gonads, where the magnitude of female-biased X-expression and female-biased expression effects associated with *flatwing* were greatest.

## (c) Feminization of the X-chromosome in gonads

As the expression of X-linked genes in gonad tissues was considerably greater in females compared with males, we tested whether ovaries-beneficial genes were overrepresented on the X-chromosome (box 1). Consistent with our third prediction, we found that genes showing ovaries-limited expression were considerably more likely than testes-limited genes to be X-linked ($\chi_1^2 = 91.19$, $p < 0.001$) (figure 2).

## 4. Discussion

Theoretical studies have proposed that the degree of sex chromosome dosage compensation will influence the prevalence and dynamics of sexual antagonism on the X chromosome, traditionally viewed as a hotspot for genomic conflict between sexes [4], by scaling effects of X-linked alleles between sexes [7,14]. We identified variable dosage compensation across tissues in *T. oceanicus*, and, consistent with the above expectation, found evidence that this variation was associated with hitherto surprising female consequences of

an X-linked, male-beneficial allele. Tissues with incomplete dosage compensation tended to exhibit female-skewed gene expression effects of the X-linked *flatwing* allele, despite the fact it exerts striking morphological consequences only in male wings. These findings support the view that, by scaling phenotypic effects of X-linked alleles, the extent of dosage compensation plays an important role in mediating the potential for sexual antagonism.

Our findings also add *T. oceanicus* to the growing list of species in which global dosage compensation is typically absent or incomplete [18,36]. At a minimum, dosage compensation in *T. oceanicus* is not uniform across tissues, as sex differences in the expression of X to autosomal genes were evident in all tissues we studied except neural tissue at 7 days post-eclosion. Early studies of dosage compensation in crickets concluded that dosage compensation was likely to be complete, either through X-inactivation (*Gryllotalpa orientalis*) or through hypertranscription of the male X (*Acheta domesticus*) [37]. However, these inferences were drawn either through indirect cytogenetic observations, or through studies of single, putatively X-linked genes. Our analysis is the first in Orthoptera to investigate dosage compensation in both a global and cross-tissue manner. Given the diversity of sex chromosome systems more generally in orthopterans [38], it would be interesting to examine whether these differences are the product of different methodologies, or whether there is true biological variation in dosage compensation mechanisms and completeness across the clade.

The cross-tissue variation in dosage compensation that we observed is an emerging theme in sex chromosome research [11,36] and has practical implications additional to its hypothetical consequences for sexual antagonism. It has been argued, for instance, that conclusions from studies of dosage compensation using whole-body gene expression data are confounded by not accounting for differences across tissues; in particular, the strong distinction in patterns of X : autosomal expression between somatic and gonad tissues which we also observed [12]. While the differences we observe across somatic tissues were smaller, they could still have consequences for studies of dosage compensation and gene expression which do not separate out tissues. Such variation could influence detectable patterns of sex-biased gene expression [11,39], particularly in organisms for which autosomal and sex-linked genes have not been identified, and, if unaccounted for, could create problems for library normalization procedures typically used in gene expression analyses [40].

Our findings support the expectation that the degree of sex chromosome dosage compensation influences the magnitude of gene expression effects of X-linked variants between the sexes (box 1). The general pattern we observe of incomplete dosage compensation is perhaps surprising given *flatwing's* male-specific fitness benefits, and the predictions laid out in box 1, which indicate incomplete dosage compensation should favour the spread of female-beneficial variants. However, in Hawaiian populations, selection for *flatwing* in males is so strong that even if there were substantial female fitness costs, these would be strongly outweighed by male benefit [22]. In other words, the extent of dosage compensation is unlikely to have played an important role in the spread of *flatwing*. It is nevertheless worth noting that developing wing tissue showed near-complete dosage compensation (figure 1; electronic supplementary material, figures S3 and S7) despite statistically significant

female-biased X-expression when compared with autosomes, for it is this tissue which ultimately gives rise to the altered wing morphology that silences males, and upon which natural selection acts in Hawaiian cricket populations.

Our observation that ovaries-limited genes were disproportionately X-linked compared with testes-limited genes highlights how female-beneficial genes might accumulate on the X [9], particularly when they are expressed in tissues with absent or incomplete dosage compensation [11,14] (box 1). However, it is difficult to disentangle cause and effect: in the crickets, do large sex differences in X-expression in gonads favour the subsequent evolution of female-beneficial genes on the X, or does the presence of female-beneficial genes on the X drive the evolution of incomplete dosage compensation to enhance their effects? Another possibility is that there are upper limits on the extent of hyper-expression of X-linked genes in males which could inhibit the spread of male-biased genes on the X [41], although these limits appear unlikely to explain the pattern we observed in gonads given the apparent lack of dosage compensating mechanisms. Fully understanding the consequences of incomplete dosage compensation for the spread of sexually antagonistic variation demands further research, for example, in a context where researchers can manipulate the expression of, and selection on, sexually antagonistic loci on the X.

Finally, while we have focussed on the consequences of sexual antagonism, other evolutionary processes may also be modulated by variable and incomplete dosage compensation. Previous theoretical and empirical work has shown that dosage compensation may alter the relative rate of evolution on the autosomes and sex chromosomes (the faster X effect) [5,42–44], with less complete dosage compensation reducing the rate of adaptive substitution on the X chromosome relative to the autosomes [5]. By contrast, certain dosage compensation mechanisms, such as X-inactivation, might increase the rate of adaptive substitution on the X [42]. Thus, in *T. oceanicus*, we should expect the strongest faster-X effect among genes expressed in neural tissue, and the weakest—if any at all—in gonads. More generally, the incomplete but variable patterns of dosage compensation, and the surprisingly large X chromosome, suggests that *T. oceanicus* and Orthoptera more widely may prove fertile ground for investigation of various aspects of sex chromosome evolution.

Data accessibility. All RNA-seq data is publicly available (accession numbers: PRJEB40088, PRJNA344019, PRJEB27211), with details of source publications provided in electronic supplementary material, table S1. R scripts are available as electronic supplementary material.

Authors' contributions. J.G.R. conceived the study and designed experiments with input from T.J.H. and N.W.B.; J.G.R. performed analyses, and all authors participated in manuscript writing led by J.G.R.

Competing interests. We declare we have no competing interests.

Funding. The study received funding from the UK Natural Environment Research Council to N.W.B. and J.G.R. grant no. (NE/T0006191/1) and NWB (NE/L011255/1, NE/I027800/1). T.J.H. was supported by a PhD studentship from the University of St Andrews School of Biology. We are grateful to the St Andrews School of Biology Research Committee for research funding support.

Acknowledgements. We gratefully acknowledge Sonia Pascoal for her role in originally collecting two of the previously published RNA-seq datasets which we used in our analysis. We are also grateful to four anonymous reviewers, the Associate Editor and colleagues in the Centre for Biological Diversity at the University of St Andrews for valuable feedback that improved our manuscript. We also thank Megan McGunnigle, Audrey Grant and Dave Forbes for their assistance with cricket rearing and maintenance.

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
