## [Peer Review File · Proceedings of the Royal Society B: Biological Sciences]

Review History

RSPB-2020-2785.R0 (Original submission)

Review form: Reviewer 1

Recommendation

Major revision is needed (please make suggestions in comments)

Scientific importance: Is the manuscript an original and important contribution to its field?

Good

General interest: Is the paper of sufficient general interest?

Good

Quality of the paper: Is the overall quality of the paper suitable?

Good

Is the length of the paper justified?

Yes

Should the paper be seen by a specialist statistical reviewer?

No

Do you have any concerns about statistical analyses in this paper? If so, please specify them explicitly in your report.

Yes

It is a condition of publication that authors make their supporting data, code and materials available - either as supplementary material or hosted in an external repository. Please rate, if applicable, the supporting data on the following criteria.

Is it accessible?

Yes

Is it clear?

Yes

Is it adequate?

Yes

Do you have any ethical concerns with this paper?

No

Comments to the Author

The authors use RNA-seq data to investigate the level of X chromosome dosage compensation in different tissues of the cricket, *Teleogryllus oceanicus*. They find that it is generally incomplete (lower expression of X in males than females), but the extent varies among tissues. By comparing flatwing phenotypes to controls, they find that there are more differentially-expressed (DE) genes in females than males, and that the number of DE genes is correlated with the extent of dosage compensation. They interpret these results as support for the hypothesis that dosage compensation mediates sexual antagonism.

Although the paper is generally clear and well-structured, I have a few concerns with the statistical analysis and the interpretation of the results:

1. line 90, "the surprising magnitude of flatwing-associated gene expression effects in females". Up to this point, the authors haven't mentioned anything about the magnitude of gene expression effects in females, other than to state that "there are recent reports that the flatwing mutation has pleiotropic or otherwise linked effects on gene expression". I assume the reader could get this information from reference 22, but otherwise it is confusing that the authors propose an explanation for something that has not been mentioned before.

2. At the end of the introduction, the authors emphasise three predictions (written in boldface). These are presented as predictions of sexual antagonism being affected by dosage compensation, but two of them (predictions 1 and 3) could also be predicted simply from what is known in other insects. For example, in *Drosophila* it is known that dosage compensation varies among tissues and is likely absent in gonads. Similarly, it is also known that the X is feminized, with an excess of ovary-specific genes. Thus, it is not such a bold prediction to predict that what has been seen in other insects will also be seen in this insect. In fact, the authors later state "*T. oceanicus* thus represents another of a growing list of species which show incomplete sex chromosome dosage compensation" (line 228).

3. line 175, "This more stringent criterion was used to check that the heightened likelihood of false positives in datasets with fewer replicates per group did not influence our results". I would think that false-negatives would be a bigger issue than false positives. When there are few replicates, you may lack the statistical power to detect DE genes. Even if the number of replicates is the same, the statistical power will also be affected by the number of mapped reads in each library. The authors don't provide information on the read numbers. One approach to address this concern would be to sub-sample the reads of the larger datasets to match the number of reads in

the smallest dataset. Similarly, the number of replicates could be reduced randomly in the data sets with more replicates. Another approach would be to use a fold-change cutoff to determine differentially expressed genes (e.g. 2-fold), as this does not depend on statistical power. I'm not suggesting that either of these approaches should replace a statistical analysis of the full data set, but they would be a good way to explore its reliability. This is especially important because the results differ depending on the FDR (fig. 2). From fig. 2, it appears that wingbud is the main (perhaps only) tissue that is sensitive to the FDR. This is concerning, as wingbud might be expected to be the most relevant tissue given the phenotype under consideration.

4. line 203, "mean expression of X-linked to autosomal genes" - wouldn't it be better to use the median here, as expression is highly variable from gene to gene and it would reduce the impact of the "extreme" genes. Does fig. 1A show means or medians?

5. line 315 and elsewhere, the authors are careful in their wording, stating that variation in dosage compensation is associated with "female consequences", but these consequences are not clear. The implication seems to be that the consequences are negative. That is, that the altered expression in females has a negative fitness impact and represents a trade-off for the positive fitness impact in males. Is there evidence for this? Is there a particularly negative effect in female gonad? It is also not clear to me that the results are specific to sexual antagonism. The authors have only tested one mutation (flatwing), which is thought to be subject to sex-specific selection. However, what would happen if they looked at another mutation that had similar (organismal) phenotypic effects in both sexes? Would it have the same effect on gene expression? I think the authors' hypothesis is that any mutation that influences gene expression should show the same pattern. If so, then why look specifically at flatwing? What would happen if, instead of using the flatwing phenotype, the replicates were randomly assigned to 2 groups and the analysis repeated. Would one get the same results (ie, more DE genes in females than males and correlation with extent of DC)?

Review form: Reviewer 2

Recommendation

Accept as is

Scientific importance: Is the manuscript an original and important contribution to its field?

Excellent

General interest: Is the paper of sufficient general interest?

Excellent

Quality of the paper: Is the overall quality of the paper suitable?

Excellent

Is the length of the paper justified?

Yes

Should the paper be seen by a specialist statistical reviewer?

No

Do you have any concerns about statistical analyses in this paper? If so, please specify them explicitly in your report.

No

It is a condition of publication that authors make their supporting data, code and materials available - either as supplementary material or hosted in an external repository. Please rate, if applicable, the supporting data on the following criteria.

Is it accessible?

Yes

Is it clear?

Yes

Is it adequate?

Yes

Do you have any ethical concerns with this paper?

No

Comments to the Author

In this ms, Rayner et al. address the question of how sexual antagonism on the sex chromosome is affected by dosage compensation. They use a very nice system in which a Hawaiian field cricket (*Teleogryllus oceananicus*) has recently evolved a X-linked mutation (flatwing) increasing male fitness and decreasing female one. They used RNA-seq data in normal wing and flatwing genotypes in each sex and across different tissues. They found that dosage compensation (DC) is incomplete in *T. oceananicus*, with the level of incompleteness varying among tissues. They found that in tissues with less complete DC, the changes in expression specific to flatwing in females were stronger. They found that the X chromosome is enriched in ovary-specific genes, the ovary being the tissue with the least DC. They conclude that DC scales phenotypic effects of X-linked genes between sexes, which supports the view that DC influences the dynamics of sexual antagonism on the X.

This study has two merits. First, it is a thorough analysis of DC in both a chromosome-wide and cross-tissues manner. It is not frequent in the literature to find studies of DC in several tissues. It is the first report of the study of DC in an orthopteran insect. Second, they have used the nice flatwing system to provide data about how DC interferes with sexual antagonism on the X. I could not detect any technical flaws, the ms is very well written. I have no comments nor suggestions for improvement, which has never happened to me before in 20 years of reviewing activity... Congratulations to the authors for this nice work.

Decision letter (RSPB-2020-2785.R0)

07-Dec-2020

Dear Mr Rayner:

I am writing to inform you that your manuscript RSPB-2020-2785 entitled "Variable dosage compensation is associated with female consequences of an X-linked, male-beneficial mutation" has, in its current form, been rejected for publication in *Proceedings B*.

This action has been taken on the advice of referees, who have recommended that substantial revisions are necessary. With this in mind we would be happy to consider a resubmission, provided the comments of the referees are fully addressed. However please note that this is not a provisional acceptance.

The resubmission will be treated as a new manuscript. However, we will approach the same reviewers if they are available and it is deemed appropriate to do so by the Editor. Please note

that resubmissions must be submitted within six months of the date of this email. In exceptional circumstances, extensions may be possible if agreed with the Editorial Office. Manuscripts submitted after this date will be automatically rejected.

Sincerely,
Dr Sasha Dall
<mailto:proceedingsb@royalsociety.org>

Associate Editor
Board Member: 1
Comments to Author:

Both reviewers were enthusiastic, although Reviewer 1 raises some very important alternative interpretations that the authors should consider. Having read the paper myself, I am also concerned that the authors have interpreted their data in terms of sexual conflict without having considered alternative interpretations. For example, there are some mechanistic considerations that need to be discussed. Specifically, one reason for the deficit of male-biased genes on X-chromosomes is not sexual conflict, but rather the limits to transcription. This was discussed by Vicoso and Charlesworth (2009 *Journal of Molecular Evolution*) as it pertains to the type of dosage compensation in *Drosophila*, but would be expected in all XY systems to some extent. Specifically, for genes that are highly expressed, it may simply not be possible to increase expression sufficiently from the single X in males to create male bias due to limitations to the transcriptional machinery. Indeed, in species with incomplete dosage compensation, the dose effect is greatest for highly expressed genes (see Harrison Mank and Wedell *GBE* 2012; Naurin, Hasselquist, Bensch, Hansson *PLOS ONE* 2012), consistent with a greater level of passive buffering at lower expression levels. It would be helpful if the authors would assess the effect of expression level on dosage compensation.

Also, with regard to Fig 1 Panel A. Specifically, I find it very difficult to accept that sexual conflict could drive X:A ratios so high in both sexes, and I suspect there is an issue with normalization. In nearly all systems assessed, female X:A ratios are roughly similar to the autosomal diploid average in females, even in gonads that are highly sexualized. It would be very surprising if crickets were the exception to this pattern, and by such a massive degree.

Reviewer(s)' Comments to Author:
Referee: 1

Comments to the Author(s)
The authors use RNA-seq data to investigate the level of X chromosome dosage compensation in different tissues of the cricket, *Teleogryllus oceanicus*. They find that it is generally incomplete (lower expression of X in males than females), but the extent varies among tissues. By comparing

flatwing phenotypes to controls, they find that there are more differentially-expressed (DE) genes in females than males, and that the number of DE genes is correlated with the extent of dosage compensation. They interpret these results as support for the hypothesis that dosage compensation mediates sexual antagonism.

Although the paper is generally clear and well-structured, I have a few concerns with the statistical analysis and the interpretation of the results:

1. line 90, "the surprising magnitude of flatwing-associated gene expression effects in females". Up to this point, the authors haven't mentioned anything about the magnitude of gene expression effects in females, other than to state that "there are recent reports that the flatwing mutation has pleiotropic or otherwise linked effects on gene expression". I assume the reader could get this information from reference 22, but otherwise it is confusing that the authors propose an explanation for something that has not been mentioned before.
2. At the end of the introduction, the authors emphasise three predictions (written in boldface). These are presented as predictions of sexual antagonism being affected by dosage compensation, but two of them (predictions 1 and 3) could also be predicted simply from what is known in other insects. For example, in *Drosophila* it is known that dosage compensation varies among tissues and is likely absent in gonads. Similarly, it is also known that the X is feminized, with an excess of ovary-specific genes. Thus, it is not such a bold prediction to predict that what has been seen in other insects will also be seen in this insect. In fact, the authors later state "T. oceanicus thus represents another of a growing list of species which show incomplete sex chromosome dosage compensation" (line 228).
3. line 175, "This more stringent criterion was used to check that the heightened likelihood of false positives in datasets with fewer replicates per group did not influence our results". I would think that false-negatives would be a bigger issue than false positives. When there are few replicates, you may lack the statistical power to detect DE genes. Even if the number of replicates is the same, the statistical power will also be affected by the number of mapped reads in each library. The authors don't provide information on the read numbers. One approach to address this concern would be to sub-sample the reads of the larger datasets to match the number of reads in the smallest dataset. Similarly, the number of replicates could be reduced randomly in the data sets with more replicates. Another approach would be to use a fold-change cutoff to determine differentially expressed genes (e.g. 2-fold), as this does not depend on statistical power. I'm not suggesting that either of these approaches should replace a statistical analysis of the full data set, but they would be a good way to explore its reliability. This is especially important because the results differ depending on the FDR (fig. 2). From fig. 2, it appears that wingbud is the main (perhaps only) tissue that is sensitive to the FDR. This is concerning, as wingbud might be expected to be the most relevant tissue given the phenotype under consideration.
4. line 203, "mean expression of X-linked to autosomal genes" - wouldn't it be better to use the median here, as expression is highly variable from gene to gene and it would reduce the impact of the "extreme" genes. Does fig. 1A show means or medians?
5. line 315 and elsewhere, the authors are careful in their wording, stating that variation in dosage compensation is associated with "female consequences", but these consequences are not clear. The implication seems to be that the consequences are negative. That is, that the altered expression in females has a negative fitness impact and represents a trade-off for the positive fitness impact in males. Is there evidence for this? Is there a particularly negative effect in female gonad? It is also not clear to me that the results are specific to sexual antagonism. The authors have only tested one mutation (flatwing), which is thought to be subject to sex-specific selection. However, what would happen if they looked at another mutation that had similar (organismal) phenotypic effects in both sexes? Would it have the same effect on gene expression? I think the authors' hypothesis is that any mutation that influences gene expression should show the same pattern. If so, then why look specifically at flatwing? What would happen if, instead of using the flatwing

phenotype, the replicates were randomly assigned to 2 groups and the analysis repeated. Would one get the same results (ie, more DE genes in females than males and correlation with extent of DC)?

Referee: 2

Comments to the Author(s)

In this ms, Rayner et al. address the question of how sexual antagonism on the sex chromosome is affected by dosage compensation. They use a very nice system in which a Hawaiian field cricket (*Teleogryllus oceananicus*) has recently evolved a X-linked mutation (flatwing) increasing male fitness and decreasing female one. They used RNA-seq data in normal wing and flatwing genotypes in each sex and across different tissues. They found that dosage compensation (DC) is incomplete in *T. oceananicus*, with the level of incompleteness varying among tissues. They found that in tissues with less complete DC, the changes in expression specific to flatwing in females were stronger. They found that the X chromosome is enriched in ovary-specific genes, the ovary being the tissue with the least DC. They conclude that DC scales phenotypic effects of X-linked genes between sexes, which supports the view that DC influences the dynamics of sexual antagonism on the X.

This study has two merits. First, it is a thorough analysis of DC in both a chromosome-wide and cross-tissues manner. It is not frequent in the literature to find studies of DC in several tissues. It is the first report of the study of DC in an orthopteran insect. Second, they have used the nice flatwing system to provide data about how DC interferes with sexual antagonism on the X. I could not detect any technical flaws, the ms is very well written. I have no comments nor suggestions for improvement, which has never happened to me before in 20 years of reviewing activity... Congratulations to the authors for this nice work.

Author's Response to Decision Letter for (RSPB-2020-2785.R0)

See Appendix A.

RSPB-2021-0355.R0

Review form: Reviewer 1

Recommendation

Accept as is

Scientific importance: Is the manuscript an original and important contribution to its field?

Excellent

General interest: Is the paper of sufficient general interest?

Good

Quality of the paper: Is the overall quality of the paper suitable?

Excellent

Is the length of the paper justified?

Yes

Should the paper be seen by a specialist statistical reviewer?

No

Do you have any concerns about statistical analyses in this paper? If so, please specify them explicitly in your report.

No

It is a condition of publication that authors make their supporting data, code and materials available - either as supplementary material or hosted in an external repository. Please rate, if applicable, the supporting data on the following criteria.

Is it accessible?

Yes

Is it clear?

Yes

Is it adequate?

Yes

Do you have any ethical concerns with this paper?

No

Comments to the Author

In this revised version of the manuscript, the authors have addressed my previous concerns adequately. I have no further comments.

Decision letter (RSPB-2021-0355.R0)

19-Feb-2021

Dear Mr Rayner

I am pleased to inform you that your Review manuscript RSPB-2021-0355 entitled "Variable dosage compensation is associated with female consequences of an X-linked, male-beneficial mutation" has been accepted for publication in Proceedings B.

The referee(s) do not recommend any further changes. Therefore, please proof-read your manuscript carefully and upload your final files for publication. Because the schedule for publication is very tight, it is a condition of publication that you submit the revised version of your manuscript within 7 days. If you do not think you will be able to meet this date please let me know immediately.

To upload your manuscript, log into <http://mc.manuscriptcentral.com/prsb> and enter your Author Centre, where you will find your manuscript title listed under "Manuscripts with Decisions." Under "Actions," click on "Create a Revision." Your manuscript number has been appended to denote a revision.

You will be unable to make your revisions on the originally submitted version of the manuscript. Instead, upload a new version through your Author Centre.

- 1) A text file of the manuscript (doc, txt, rtf or tex), including the references, tables (including captions) and figure captions. Please remove any tracked changes from the text before submission. PDF files are not an accepted format for the "Main Document".
- 2) A separate electronic file of each figure (tiff, EPS or print-quality PDF preferred). The format should be produced directly from original creation package, or original software format. Please note that PowerPoint files are not accepted.
- 3) Electronic supplementary material: this should be contained in a separate file from the main text and the file name should contain the author's name and journal name, e.g
 authurname_procb_ESM_figures.pdf
 All supplementary materials accompanying an accepted article will be treated as in their final form. They will be published alongside the paper on the journal website and posted on the online figshare repository. Files on figshare will be made available approximately one week before the accompanying article so that the supplementary material can be attributed a unique DOI. Please see: <https://royalsociety.org/journals/authors/author-guidelines/>

4) Data-Sharing and data citation

It is a condition of publication that data supporting your paper are made available. Data should be made available either in the electronic supplementary material or through an appropriate repository. Details of how to access data should be included in your paper. Please see <https://royalsociety.org/journals/ethics-policies/data-sharing-mining/> for more details.

<http://datadryad.org/submit?journalID=RSPB&manu=RSPB-2021-0355> which will take you to your unique entry in the Dryad repository.

Once again, thank you for submitting your manuscript to Proceedings B and I look forward to receiving your final version. If you have any questions at all, please do not hesitate to get in touch.

Sincerely,

Dr Sasha Dall

Associate Editor

Board Member

Comments to Author:

The reviewers have done a great job in their revision, and I am happy to recommend acceptance.

Reviewer(s)' Comments to Author:

Referee: 1

Comments to the Author(s).

In this revised version of the manuscript, the authors have addressed my previous concerns adequately. I have no further comments.

Sincerely,

Proceedings B

Decision letter (RSPB-2021-0355.R1)

01-Mar-2021

Dear Mr Rayner

I am pleased to inform you that your manuscript entitled "Variable dosage compensation is associated with female consequences of an X-linked, male-beneficial mutation" has been accepted for publication in Proceedings B.

Your article has been estimated as being 8 pages long. Our Production Office will be able to confirm the exact length at proof stage.

Open Access

You are invited to opt for Open Access, making your freely available to all as soon as it is ready for publication under a CCBY licence. Our article processing charge for Open Access is £1700. Corresponding authors from member institutions (<http://royalsocietypublishing.org/site/librarians/allmembers.xhtml>) receive a 25% discount to these charges. For more information please visit <http://royalsocietypublishing.org/open-access>.

Paper charges

Sincerely,
Proceedings B
<mailto:proceedingsb@royalsociety.org>

Appendix A

Dear Editor,

We are pleased to submit a considerably revised and improved version of our previous submission, RSPB-2020-2785 (“Variable dosage compensation...”).

Our study tests an important assumption of models of sexual antagonism on the X: that the degree of dosage compensation scales phenotypic effects of X-linked variants. Our findings supported this model. We found that incomplete sex chromosome dosage compensation in crickets could explain a surprising pattern of large gene expression effects in females of an X-linked mutation, despite the fact that the mutation is under selection for male-specific fitness benefits.

We were grateful for the opportunity to respond to the insightful feedback on our manuscript. Both reviewers were enthusiastic about our study, but Reviewer 1 expressed some concerns about our analyses, and you expressed reservations about the degree of X-biased expression (cf. autosomes).

In response to these concerns, we completely revised our analysis of dosage compensation using a non-parametric approach with a strong precedent in the literature which is more robust to outlier genes with particularly high expression. While the results provide interpretation consistent with those of our previous submission, the revised analysis addresses the potential issue of outliers raised by Reviewer 1, and also places the extent of X-biased expression into a more realistic biological context.

We made various further improvements to our manuscript based on the reviewers’ comments, which are detailed in the response letter below and highlighted in our revised submission. We hope you will agree that our modifications address the concerns raised and represent a substantial improvement on our previous submission. Thank you for your continued interest in our work.

Yours sincerely,
Jack Rayner, Thomas Hitchcock, Nathan Bailey

Associate Editor

Both reviewers were enthusiastic, although Reviewer 1 raises some very important alternative interpretations that the authors should consider. Having read the paper myself, I am also concerned that the authors have interpreted their data in terms of sexual conflict without having considered alternative interpretations. For example, there are some mechanistic considerations that need to be discussed. Specifically, one reason for the deficit of male-biased genes on X-chromosomes is not sexual conflict, but rather the limits to transcription. This was discussed by Vicoso and Charlesworth (2009 Journal of Molecular Evolution) as it pertains to the type of dosage compensation in Drosophila, but would be expected in all XY systems to some extent. Specifically, for genes that are highly expressed, it may simply not be possible to increase expression sufficiently from the single X in males to create male bias due to limitations to the transcriptional machinery. Indeed, in species with incomplete dosage compensation, the dose effect is greatest for highly expressed genes (see Harrison Mank and Wedell GBE 2012; Naurin, Hasselquist, Bensch, Hansson PLOS ONE 2012), consistent with a greater level of passive buffering at lower expression levels. It would be helpful if the authors would assess the effect of expression level on dosage compensation.

Thank you for your time and effort in reading our paper and offering helpful feedback.

Firstly, while we agree that potential limits to hyperexpression of the X in males are an important consideration, we do not think these are likely to influence our result with respect to the deficit of male-limited genes present on the X chromosome. From Vicoso & Charlesworth (2009), we understand that hyper-expression of the single X in males, i.e., dosage compensation, could impose limits on further up-regulation of expression of X-linked genes in males. However, in gonads – the only tissue in which we perform this comparison of the proportion of male/female-biased genes situated on the X/autosomes – we found no evidence of dosage compensation, as males expressed X-linked genes at about half the level of females. This is consistent with other work finding little or no dosage compensation in gonad tissues. Therefore, we believe differences in X ploidy combined with incomplete dosage compensation are the most likely explanation for the paucity of testes-limited genes on the X. We have included discussion of this topic at line 381.

Secondly, following our revised analysis (described below), we also tested whether patterns of dosage compensation differed across expression levels, as suggested above. Among the four tissues which reported incomplete dosage compensation, three indeed showed a trend for greater X-dose effect at higher average expression levels (Fig. S1, mentioned on line 253 of revised MS).

Also, with regard to Fig 1 Panel A. Specifically, I find it very difficult to accept that sexual conflict could drive X:A ratios so high in both sexes, and I suspect there is an issue with normalization. In nearly all systems assessed, female X:A ratios are roughly similar to the autosomal diploid average in females, even in gonads that are highly sexualized. It would be very surprising if crickets were the exception to this pattern, and by such a massive degree.

We agree with the above point, and we are grateful that you brought it to our attention. Following further investigation, we found that high representation of X-linked genes among

the most highly expressed genes influenced our results regarding the high average expression of X-linked genes in some tissues. This is a shortcoming of the previous method we used to assess dosage compensation, which was based on average expression and thus susceptible to outliers (as Reviewer 1 suggested). We have therefore reformed this analysis using each of the two commonly used non-parametric approaches for quantifying dosage compensation described in the review of Gu & Walters (2017, GBE). These are described on line 205 of our revised Methods, and the re-analysed results are reported throughout the revised Results section and Supporting Information. The revised results do indicate that the X chromosome is expressed more highly than autosomal genes, which appears to result from low median expression in some autosomal linkage groups (Figs. S1, S2), but the magnitude of this effect is much smaller than in the previous analysis: across the 5 tissues, the average ratio of median expression of X/A genes is 1.3. Besides providing a more realistic view of the high expression of the X chromosome in general, our revised analysis, which is more robust to outliers, does not qualitatively change our results or their biological interpretation.

Referee: 1

*The authors use RNA-seq data to investigate the level of X chromosome dosage compensation in different tissues of the cricket, *Teleogryllus oceanicus*. They find that it is generally incomplete (lower expression of X in males than females), but the extent varies among tissues. By comparing flatwing phenotypes to controls, they find that there are more differentially-expressed (DE) genes in females than males, and that the number of DE genes is correlated with the extent of dosage compensation. They interpret these results as support for the hypothesis that dosage compensation mediates sexual antagonism.*

Although the paper is generally clear and well-structured, I have a few concerns with the statistical analysis and the interpretation of the results:

Thank you for your valuable time and feedback.

1. line 90, "the surprising magnitude of flatwing-associated gene expression effects in females". Up to this point, the authors haven't mentioned anything about the magnitude of gene expression effects in females, other than to state that "there are recent reports that the flatwing mutation has pleiotropic or otherwise linked effects on gene expression". I assume the reader could get this information from reference 22, but otherwise it is confusing that the authors propose an explanation for something that has not been mentioned before.

We agree it is useful to make these clearer, so have added further explanation on line 80. The sentence now reads:

"Recent reports indicate that the *flatwing* mutation has pleiotropic or otherwise linked consequences for female gene expression [22], and for female life history traits (reduced reproductive investment, increased rate of mating failure, increased somatic mass index, growth rate) [23–25]."

2. At the end of the introduction, the authors emphasise three predictions (written in boldface). These are presented as predictions of sexual antagonism being affected by dosage compensation, but two of them (predictions 1 and 3) could also be predicted simply from what is known in other insects. For example, in *Drosophila* it is known that dosage compensation varies among tissues and is likely absent in gonads. Similarly, it is also known that the X is feminized, with an excess of ovary-specific genes. Thus, it is not such a bold prediction to predict that what has been seen in other insects will also be seen in this insect. In fact, the authors later state "*T. oceanicus* thus represents another of a growing list of species which show incomplete sex chromosome dosage compensation" (line 228).

Empirical patterns from other taxa are interesting, and we cite relevant examples in our predictions at lines 111 and 121, as well as in the discussion quoted above. Our study's primary aim was to test support for an assumption underpinning models of sexual antagonism, and it is therefore to be expected that our predictions are intuitive given the available evidence in other taxa. While we do not dispute that our results regarding dosage compensation and X-feminization have precedent in other species, *T. oceanicus* offers a useful ecological and evolutionary context in which to study these features and relate them

to female consequences of an X-linked mutation.

3. line 175, "This more stringent criterion was used to check that the heightened likelihood of false positives in datasets with fewer replicates per group did not influence our results". I would think that false-negatives would be a bigger issue than false positives. When there are few replicates, you may lack the statistical power to detect DE genes. Even if the number of replicates is the same, the statistical power will also be affected by the number of mapped reads in each library. The authors don't provide information on the read numbers. One approach to address this concern would be to sub-sample the reads of the larger datasets to match the number of reads in the smallest dataset. Similarly, the number of replicates could be reduced randomly in the data sets with more replicates. Another approach would be to use a fold-change cutoff to determine differentially expressed genes (e.g. 2-fold), as this does not depend on statistical power. I'm not suggesting that either of these approaches should replace a statistical analysis of the full data set, but they would be a good way to explore its reliability. This is especially important because the results differ depending on the FDR (fig. 2). From fig. 2, it appears that wingbud is the main (perhaps only) tissue that is sensitive to the FDR. This is concerning, as wingbud might be expected to be the most relevant tissue given the phenotype under consideration.

In assessing differential expression we do not combine different datasets, which will indeed differ in numbers of mapped reads. Rather, we perform differential expression analysis separately for each of the tissues/datasets and compare numbers of DE genes between sexes within each of these tissues. It is only the difference between sexes in numbers of DE genes which is compared across tissues, therefore any differences between sexes in the number of mapped reads are accounted for by the analysis in EdgeR, which normalises library sizes according to number of mapped reads and other features. We therefore do not think that the number of mapped reads will influence our results. However, we do agree that the issue of differences in replicate size is certainly worth investigating.

To investigate whether replicate size could have influenced our results, we subsampled replicates from the neural_7d (neural tissue at 7-days post-eclosion) dataset, which had more replicates than the other tissues. To do so, we randomly selected 3 samples from each of the replicates (to equalize replicate size with that of the other tissues), and performed differential expression analysis using those 3 samples rather than the full 6. We repeated this subsampling process 20 times and then compared the number of genes reported as DE between genotypes in each of the sexes, across all of the iterations. We used a paired Wilcoxon rank-sum test to look for differences in the number of DE genes reported for males and females in each iteration, and found there was no evidence of a difference at either $FDR < 0.05$ or $FDR < 0.01$ thresholds – supporting our original results. This is now described at line 179 in methods, line 276 in results, with further detail in the Supporting Information.

With regard to using log-fold change thresholds rather than, or in addition to, significance thresholds, we believe that these are more likely to introduce noise to our results than reduce it. In a dataset with relatively few replicates per group (i.e., 3 per genotype), it is likely that a substantial number of genes will show log-fold differences in their mean

expression between groups by chance, and the FDR-based approach is designed to filter these out.

In response to the concern raised that the wingbud tissue is most sensitive to differences in FDR threshold, we suggest that this is likely due to low number of DE genes in this tissue at either threshold. We now mention this on line 282.

4. line 203, "mean expression of X-linked to autosomal genes" - wouldn't it be better to use the median here, as expression is highly variable from gene to gene and it would reduce the impact of the "extreme" genes. Does fig. 1A show means or medians?

Fig. 1A previously showed means; apologies, this was not clear. We previously considered mean values useful as they are proportional to the sum of expression from X/A chromosomes. However, we agree with the above point that these averages are likely skewed by genes with extreme expression. As discussed in the response to the Editor's comments above, we re-performed the analysis of dosage compensation by calculating gene-wise ratios of female:male expression, then compared distributions of these ratios between X-linked and autosomal gene groups. This pair-wise, non-parametric approach has a strong precedent in studies of dosage compensation (described and cited at line 204), and ensures outliers will not influence our results. Thus, in quantifying the degree of dosage compensation, we now compare median F:M ratios between X-linked and autosomal gene sets, as suggested. Reassuringly, the results produce interpretation consistent with our previous findings – see updated Fig. 1 and Fig. S1.

5. line 315 and elsewhere, the authors are careful in their wording, stating that variation in dosage compensation is associated with "female consequences", but these consequences are not clear. The implication seems to be that the consequences are negative. That is, that the altered expression in females has a negative fitness impact and represents a trade-off for the positive fitness impact in males. Is there evidence for this? Is there a particularly negative effect in female gonad? It is also not clear to me that the results are specific to sexual antagonism. The authors have only tested one mutation (flatwing), which is thought to be subject to sex-specific selection. However, what would happen if they looked at another mutation that had similar (organismal) phenotypic effects in both sexes? Would it have the same effect on gene expression? I think the authors' hypothesis is that any mutation that influences gene expression should show the same pattern. If so, then why look specifically at flatwing? What would happen if, instead of using the flatwing phenotype, the replicates were randomly assigned to 2 groups and the analysis repeated. Would one get the same results (ie, more DE genes in females than males and correlation with extent of DC)?

You are correct that we do not make a direct link between the extent of these consequences of female gene expression and fitness. This is an important distinction that we now make clearer at line 86. Our hypothesis was that dosage compensation would influence the magnitude of genome-wide, gene expression consequences caused by the presence of the specific X-linked allele, *flatwing*. If dosage compensation were complete, then we anticipated that males and females should exhibit similar, or male-biased (given the male-limited effects on wing morphology) *flatwing*-associated consequences. In contrast, if

dosage compensation were absent or incomplete, then greater consequences might be expected in females. It is in this context that our results have implications for study of sexual antagonism – specifically, testing the assumption of models of sexual antagonism that dosage compensation should be an important parameter in predicting conflict on the X, by scaling phenotypic effects between sexes. We mention on line 367 that it is unlikely dosage compensation played an important role in the evolutionary dynamics of *flatwing* itself.

As you suggest, our expectation is that a similar pattern should be observed for any X-linked, additive mutation. However, while the suggestion to investigate this in other genes is without doubt an interesting proposal, the experimental design only allows us to test this in the context of *flatwing*, for which we have purebreeding lines for the two respective genotypes. It is highly unlikely other X-linked mutations segregate in a consistent manner across tissues and datasets within these lines (especially as females would need to be homozygous), and, in addition, we consider that RNA-seq data with a limited number of samples per tissue would be insufficient to perform the above test.

Referee: 2

Comments to the Author(s)

*In this ms, Rayner et al. address the question of how sexual antagonism on the sex chromosome is affected by dosage compensation. They use a very nice system in which a Hawaiian field cricket (*Teleogryllus oceananicus*) has recently evolved a X-linked mutation (flatwing) increasing male fitness and decreasing female one. They used RNA-seq data in normal wing and flatwing genotypes in each sex and across different tissues. They found that dosage compensation (DC) is incomplete in *T. oceananicus*, with the level of incompleteness varying among tissues. They found that in tissues with less complete DC, the changes in expression specific to flatwing in females were stronger. They found that the X chromosome is enriched in ovary-specific genes, the ovary being the tissue with the least DC. They conclude that DC scales phenotypic effects of X-linked genes between sexes, which supports the view that DC influences the dynamics of sexual antagonism on the X. This study has two merits. First, it is a thorough analysis of DC in both a chromosome-wide and cross-tissues manner. It is not frequent in the literature to find studies of DC in several tissues. It is the first report of the study of DC in an orthopteran insect. Second, they have used the nice flatwing system to provide data about how DC interferes with sexual antagonism on the X. I could not detect any technical flaws, the ms is very well written. I have no comments nor suggestions for improvement, which has never happened to me before in 20 years of reviewing activity... Congratulations to the authors for this nice work.*

Thank you for the kind comments. We are very pleased that you found our study to be of merit.